# Current Advances in the Treatment of BRAF-Mutant Melanoma

**DOI:** 10.3390/cancers12020482

**Published:** 2020-02-19

**Authors:** Hima Patel, Nour Yacoub, Rosalin Mishra, Aaron White, Long Yuan, Samar Alanazi, Joan T. Garrett

**Affiliations:** 1James. L. Winkle College of Pharmacy, University of Cincinnati, Cincinnati, OH 45243, USA; patel2h2@mail.uc.edu (H.P.); mishrarn@ucmail.uc.edu (R.M.); yuanlg@mail.uc.edu (L.Y.); alanazsa@mail.uc.edu (S.A.); 2College of Medicine, Northeast Ohio Medical University, Rootstown, OH 44274, USA; nyacoub@neomed.edu; 3College of Medicine, University of Cincinnati, Cincinnati, OH 45243, USA; white3ai@mail.uc.edu

**Keywords:** melanoma, metastatic, resistance, BRAF, immunotherapy

## Abstract

Melanoma is the most lethal form of skin cancer. Melanoma is usually curable with surgery if detected early, however, treatment options for patients with metastatic melanoma are limited and the five-year survival rate for metastatic melanoma had been 15–20% before the advent of immunotherapy. Treatment with immune checkpoint inhibitors has increased long-term survival outcomes in patients with advanced melanoma to as high as 50% although individual response can vary greatly. A mutation within the MAPK pathway leads to uncontrollable growth and ultimately develops into cancer. The most common driver mutation that leads to this characteristic overactivation in the MAPK pathway is the B-RAF mutation. Current combinations of BRAF and MEK inhibitors that have demonstrated improved patient outcomes include dabrafenib with trametinib, vemurafenib with cobimetinib or encorafenib with binimetinib. Treatment with BRAF and MEK inhibitors has met challenges as patient responses began to drop due to the development of resistance to these inhibitors which paved the way for development of immunotherapies and other small molecule inhibitor approaches to address this. Resistance to these inhibitors continues to push the need to expand our understanding of novel mechanisms of resistance associated with treatment therapies. This review focuses on the current landscape of how resistance occurs with the chronic use of BRAF and MEK inhibitors in BRAF-mutant melanoma and progress made in the fields of immunotherapies and other small molecules when used alone or in combination with BRAF and MEK inhibitors to delay or circumvent the onset of resistance for patients with stage III/IV BRAF mutant melanoma.

## 1. Introduction

Melanoma is the uncontrollable division of melanocytes located within the deep layer of the epidermis [1]. Although invasive melanoma is the third most common type of skin cancer, it is the most serious compared to its other two counterparts- basal cell carcinoma and squamous cell carcinoma. The American Cancer Society estimates that in 2019 there will be 96,480 new cases of melanoma diagnosed accompanied by 7,230 expected deaths. The five-year survival rate for metastatic melanoma has been 15–20% [2], although these statistics are rapidly improving with the success of immune checkpoint inhibitors. Treatment with immune checkpoint inhibitors demonstrated substantial clinical efficacy along with long-term survival outcomes in patients with advanced melanoma [3,4,5]. There are several clinical trials such as KEYNOTE-002, CheckMate067 and CheckMate064, which validate these findings, as detailed in Table 1.

An important risk factor for melanoma is exposure to direct and chronic ultraviolet (UV) radiation which, increases the risk for DNA damage and leads to genetic changes. Familial history is also one of the risk factors for developing melanoma, with cyclin-dependent kinase inhibitor 2A (CDKN2A) and cyclin-dependent kinase 4 (CDK4) being the most common heritable mutations [1]. Germ-line polymorphisms of the melanocortin-1 receptor (MC1R) also confers susceptibility due to its ability to control the level of skin pigmentation in response to UV radiation [2,6]. 

The molecular changes occurring in the progression of melanoma serve as points for therapeutic interference. Typically, the preliminary change of a melanocyte into a benign nevus remains controlled and is non-cancerous. However, some molecular changes can lead to overactivation of growth regulatory pathways, such as the mitogen-activated protein kinase (MAPK) signaling pathway [2]. The MAPK pathway is crucial in relaying extracellular signals in order to keep a balance of growth/proliferation and apoptosis within the cell. A mutation within MAPK pathway leads to uncontrollable growth and ultimately develops into cancer [7]. The most common driver mutation that leads to this characteristic overactivation in the MAPK pathway is the B-RAF mutation [2]. Raf is a family of oncogenic serine-threonine protein kinases within the MAPK pathway with three isoforms: A-RAF, B-RAF, and C-RAF [7,8]. B-RAF-mutant melanoma accounts for nearly 50% of metastatic melanoma cases. Substitution of valine (V) for glutamic acid (E) at amino acid position 600 (V600E) represents 84.6% of the B-RAF mutations. A second common substitution of valine (V) for lysine (K) at amino acid position 600 (V600K), representating 7.7% of the B-RAF mutations [9]. B-RAF mutant melanoma is typically found in younger patients and is characterized by a superficial spreading tumor or nodular tumor that can be found in areas without chronic sun exposure. It has a higher chance of metastasizing into brain along with a shorter survival time as compared to the non-BRAF mutant melanoma. A B-RAF mutation alone may not contribute to the development of melanoma; but accompanying driver mutations in tumor suppressor genes are commonly indispensable leading to the development of malignant melanoma [9].

Several studies have addressed the need for molecular testing with respect to B-RAF mutations in order to tailor the best course of treatment available for each patient. [9,10]. Treatment options include surgery, immunotherapy, targeted therapy, chemotherapy, inclusion in a clinical trial and radiation [11]. Stage I and II melanoma can typically be surgically excised in concert with a sentinel lymph node biopsy if there is concern for metastasis. Stage III and IV melanoma require more systemic interventions due to the aggressiveness of the tumor and increased tumor burden [8]. The current standard of care for a patient diagnosed with B-RAF mutant metastatic melanoma is to first consider eligibility for a metastasectomy. Regardless of qualifications for the metastasectomy, the next step is to design a treatment regimen with either checkpoint inhibition immunotherapy or a molecularly targeted therapy [9,10]. 

Current targeted therapies include a combination of B-RAF and MEK inhibitors. Vemurafenib was the first FDA-approved B-RAF inhibitor in 2011, followed by approval of dabrafenib in 2013 [8]. The most recent FDA-approved B-RAF inhibitor is encorafenib, approved in 2018 [12]. In parallel with the discovery and use of B-RAF inhibitors opened up the avenue for development of MEK inhibitors, targeting a molecule downstream of the B-RAF protein. The first MEK-inhibitor, trametinib, was approved by FDA in 2013, followed by approval of cobimetinib in 2015 [8]. Another recently FDA-approved MEK inhibitor was binimetinib [12]. Current combinations of BRAF and MEK inhibitors that have demonstrated improved patient outcomes include dabrafenib with trametinib, vemurafenib with cobimetinib or encorafenib with binimetinib [13]. Treatment with BRAF and MEK inhibitors has met with some challenges as patient responses began to drop due to the development of resistance to these inhibitors which paved the way for development of immunotherapies and other small molecule inhibitor approaches to address this. 

Current immunotherapies include the anti-cytotoxic T-lymphocyte antigen 4 antibody (anti-CTLA-4) and two anti-programmed death protein 1 antibodies (anti-PD1) [8]. Current preclinical and clinical trials are underway to determine the efficacy and benefits of combining immunotherapy treatment regimen alone or in combination with BRAF and MEK inhibitors for treatment of patients with BRAF mutant melanoma [13,14,15,16]. New avenues exploring the possible combination therapies of BRAF/MEK inhibitors with immunotherapy drugs are being tested. Combination therapies are not only limited to MAPK pathway targeted therapies plus immunotherapy but have expanded to include other molecules such as AXL and ROS that have a role in the development of drug resistance. These have emerged as alternative treatment options for treating metastatic melanoma patients. Preclinical and clinical trials evaluating the efficacy of various PI3K and CDK4/6 inhibitors in combination with BRAF and MEK inhibitors are also initiated [17,18,19,20,21,22]. 

This review focuses on the current landscape of resistance with the chronic use of BRAF and MEK inhibitors in BRAF -mutant melanoma and progress made in the fields of immunotherapies and other small molecules when used alone or in combination with BRAF and MEK inhibitors to delay or circumvent the onset of resistance for patients with stage III/IV melanoma. 

## 2. Mechanisms of Resistance 

The development and use of the BRAF targeted inhibitors, Vemurafenib and dabrafenib, has improved the treatment arena for patients with metastatic melanoma. However, over half of patients treated with these single agent inhibitors begin to show signs of tumor recurrence within 6 to 7 months of treatment [23,24]. Several mechanisms of drug resistance have been proposed. There are two general types of resistance—primary resistance/intrinsic resistance and secondary or acquired resistance. Intrinsic resistance refers to those patients who do not respond to any type of BRAF inhibitor therapy and accounts for approximately 15% of patients [25]. Acquired resistance, refers to those patients who show tumor regression after an initial positive response to therapy and this is commonly observed in most melanoma patients [26]. 

### 2.1. Intrinsic Resistance 

PTEN is a negative regulator of phosphoinositde 3-kinase (PI3K) and loss of PTEN can lead to an upregulation of the PI3K/Akt pathway whose activation can explain tumor resistance [27,28]. Loss of PTEN alone does not confer a resistance state; it is typically accompanied with Akt phosphorylation and activation [26,29]. Cyclin D1 amplification (CCND1) is observed in about 15–20% of all BRAF-mutant melanoma [30,31]. CCND1 alone can accelerate the resistance in BRAF-mutant melanoma and is intensified when there is both cyclin D1 overexpression along with a cyclin dependent kinase-4 (CDK4) mutation [32]. 

Neurofibromin-1 (NF1) is a tumor suppressor and a negative regulator of RAS. Loss of NF1 is typically seen in 14% of melanoma cases and leads to activation of RAS and other downstream pathways including the MAPK and PI3K-Akt [26,33]. Loss of NF1 can also mediate resistance to RAF and MEK inhibitors [34]. 

RAC1 is part of the Rho family of small GTP binding proteins. Mutations in RAC1 are found in 4–9% of patients and is the third “hotspot” mutation in melanoma, following BRAF and NRAS [35,36,37]. RAC1 mutation status is being considered a biomarker for resistance to RAF inhibitor therapy [35]. It has been shown that pre-existing mutations in MEK1 can confer resistance to RAF inhibitor therapy [38]. 

### 2.2. Acquired Resistance 

Starting from the cell surface, several receptor tyrosine kinases (RTK) converge onto parallel pathways such as the MAPK and the PI3K-Akt pathway [39]. Upregulation of RTKs has been shown to directly activate the MAPK pathway via RAS activation [30]. Additionally, upregulation of specific RTKs such as the insulin like growth factor 1 receptor (IGFR1), platelet derived growth factor receptor β (PDGFRβ) can activate the PI3K-Akt pathway in a non-ERK dependent manner [40,41]. Epigenetic changes affecting epidermal growth factor receptor (EGFR) have been shown to also induce the PI3K-Akt pathway in melanoma resistant cells [42]. Dual activation of these pathways strongly contributes to drug resistance, as these pathways promote cell survival and proliferation. 

NRAS activating mutations are present in approximately 15–20% of melanomas [43]. Q61 mutations in NRAS keeps it constitutively active in the ‘RAS-GTP’ state [44]. The activated mutant NRAS can activate the MAPK pathway via induction of dimerization of CRAF and BRAF [45]. 

Treatment with BRAF inhibitors will only inhibit the mutant monomer in BRAF mutant melanoma cells [46]. This plays an important role in the ability for these cells to maintain RAF dimerization which in turn keeps MAPK signaling active. A phenomenon known as the ‘BRAF-inhibitor Paradox’ describes the event in which the BRAF inhibitor blocks MAPK signaling in mutant cells but activates the MAPK pathway in non-mutant cells by allowing the drug-free RAF protein to be transactivated and dimerize [47]. RAF dimerization can be fulfilled through a variety of mechanisms including alternative BRAF splicing, amplification of BRAF, and expression of different RAF isoforms such as CRAF overexpression [48,49,50,51]. Resistance to RAF inhibition include activation of HGF and its receptor MET which lead to the reactivation of the MAPK and PI3K-AKT pathways [52]. Screening for the presence of RAF inhibitor resistance genes, found in a high percentage of patients can help improve treatment outcomes especially when used in conjunction with appropriate therapeutic combinations [53].

Secondary mutations in both MEK1 and MEK2 have also been linked to acquired resistance in melanoma cell lines [54]. The resistance to MEK inhibitors is attributed to mutations in MEK1/2 which lead to constitutive activation of MEK1/2 or a mutation in the drug binding site [55]. MAPK reactivation occurs via secondary mutations in MEK1 (Q56P or E203K) which help reactive the MAPK pathway downstream [56]. Additionally, BRAF amplification along with KRAS mutations can be contributing factors to MEK1/2 inhibitor resistance [57].

The RTK AXL has also been identified as a player in both intrinsic and acquired resistance. Patients relapsed of BRAF and MEK inhibitors overexpress AXL as an adaptive response [58]. Non-genomic mechanism of acquired resistance include high expression of transcriptomic alterations and intra-tumoral immunity which involves cytolytic T-cell inflammation [59].

Combination therapy using BRAF and MEK inhibitors has also shown signs of resistance. Proposed mechanisms of resistance include BRAF gene amplification, BRAF splice-variants and mutations in MEK2 [60]. Resistance to BRAF and MEK inhibitors exists by combining or augmenting the mechanisms related to single agent BRAF inhibitor resistance. The overexpressed BRAFV600E and MEK mutants interact via the regulatory interface of BRAFV600E, R662 [61]. Acquired resistance to combination targeted therapy has other factors contributing to it, including whether the patient had received inhibitor monotherapy prior to the combination therapy or is ‘monotherapy naïve’ [62].

Many recent advancements in the treatment of metastatic melanoma have been in the field of immunotherapy. There are proposed mechanisms of resistance to BRAF and MEK inhibitor therapy involving immune system molecules. Cancer cells, in general, aim to avoid immune recognition by downregulating surface receptors that participate in co-activation of T cells as well as upregulating negative feedback pathways, such as the immune checkpoint inhibitor receptors programmed cell death protein (PD-1) and T-lymphocyte associated protein-4 (CTLA-4). PD-1 and CTLA-4 are localized on the T-cells [63]. It is documented that after 2 weeks of treatment with BRAF and MEK inhibitors, melanoma cells have been able to downregulate melanoma differentiation antigens (MDA) surface expression, decrease T cell activity, and surface display of increase immune checkpoint inhibitory receptors [64,65]. This manipulation of key immune system regulators gives melanoma cells yet another way to bypass drug resistance. The rationale to combine an immune checkpoint inhibitor therapy with targeted therapy is that the treatment with a BRAF and MEK inhibitor renders the tumor microenvironment more immunoresponsive [66]. 

The sequence of treatment for a patient with targeted therapy and immune therapy is not well established. Starting a patient on a BRAF inhibitor or anti-PD1 inhibitor is effective regardless of the treatment order, but more randomized controlled trials are required to address and establish the superiority and sequencing of one therapy over the other [67,68]. Studies have also shown that the efficacy of immunotherapy is improved in previously untreated patients compared to patients who have single agent immunotherapy failure or failure to targeted therapy [69]. Despite showing exceptional clinical efficacy, treatment with immune checkpoint inhibitors has met some difficulties with respect to development of innate and acquired resistance [70,71]. Various clinical trials evaluating immune therapies such as Toll-like receptor 9 (TRL9) agonists, neoantigen vaccines and oncoloytic viruses alone or in combination with immune checkpoint inhibitors are underway. Combination of these therapies may help combat resistance to immune checkpoint inhibitors [72,73,74,75,76,77]. 

## 3. Therapies 

Figure 1 summarizes the therapies in pre-clinical and clinical phases, described in this review to treat patients with metastatic melanoma.

## 4. Anti-PD-1/PD-L1

The immunogenic nature of melanoma was utilized to develop several immunotherapeutic treatment strategies especially with regards to the programmed cell death (PD-1) receptor and its ligand, PD-L1. Antibodies targeting the PD-1 axis has shown significant promise in the clinic for treatment of metastatic melanoma either as a monotherapy or in combination with Ipilimumab. There are several ongoing clinical trials using anti-PD1 and anti-PD-L1 antibodies. Programmed cell death protein 1 (PD-1) is an inhibitory receptor expressed on the surface of the cancer cells that inhibits the immune system via suppressing the T-cell activity. Anti-PD-1 monoclonal antibodies block the PD-1 receptor which maintains T-cells in activated state to suppress the tumor growth [78]. There are several anti-PD-1/PD-L1 monoclonal antibodies including pembrolizumab (Keytruda^®^), nivolumab (Opdivo^®^), avelumab (Bavencio^®^), durvalumab (Imfinzi^®^), cemiplimab (Libtayo^®^), atezolizumab (Tecentriq^®^), cosibelimab and INBRX-105 in several stages of clinical trial in melanoma. Pembrolizumab, nivolumab and nivolumab in combination with iIpilimumab (anti-CTLA-4 inhibitor) have been approved by FDA for treatment of melanoma.

### 4.1. Pembrolizumab/Lambrolizumab/MK-3475/SCH 900475/Keytruda

This is a humanized monoclonal antibody targeting the PD1 receptor in the lymphocytes. It was developed by Merck and approved for treatment of metastatic melanoma in 2017 [79]. 

### 4.2. Nivolumab/ONO-4538/BMS-936558/MDX1106/Opdivo

This is a human IgG4 anti-PD-1 monoclonal antibody. Nivolumab works as a checkpoint inhibitor that inhibits T-cell activation. [80]. It was developed by Medarex and Ono Pharmaceutical, and is marketed by Bristol-Myers Squibb (BMS) and Ono. Nivolumab was approved by the FDA for melanoma in 2014 [81,82]. 

### 4.3. Avelumab/MSB0010718C/Bavencio

This is a humanized monoclonal antibody developed by Merck and Pfizer that targets the PD-L1. It has been approved by FDA for treatment Merkel-cell carcinoma, an aggressive type of skin cancer [83]. It blocks the PD-1/PD-L1 receptor/ligand complex formation leading to suppression of CD8+ T cells action [84]. There is a current clinical trial (NCT01772004) investigating the safety, tolerability, pharmacokinetics and clinical activity of avelumab in melanoma [85].

### 4.4. Durvalumab/MEDI4736/Imfinzi

This is monoclonal antibody that blocks the interaction of PD-L1 with the PD-1 and CD80 (B7.1) molecules developed by Medimmune/AstraZeneca [86,87]. A phase I clinical trial (NCT02586987) is ongoing evaluating the safety and efficacy of selumetinib (AZD6244 Hyd-sulfate) in combination with durvalumab (MEDI4736) along with tremelimumab in patients with advanced solid tumours, including melanoma [88]. 

### 4.5. Atezolizumab/MPDL3280A/Tecentriq

This is a fully humanized engineered monoclonal antibody of IgG1 isotype against PD-L1 developed by Genentech [89,90]. There is an active ongoing phase II trial (NCT02303951) which includes the combination of vemurafenib, cobimetinib and atezolizumab in stage III/IV advanced melanoma patients [91]. Another phase III study (NCT02908672) compares the efficacy of atezolizumab in combination with cobimetinib and vemurafenib versus placebo control plus cobimetinib and vemurafenib in unresectable and advanced melanoma patients with BRAFV600 mutation [92]. 

### 4.6. Spartalizumab/PDR001

Spartalizumab (PDR001) is a humanized monoclonal antibody against the negative immuno-regulatory human cell surface receptor programmed death-1 (PD-1, PCD-1) was developed by Novartis. This suppresses T-cell activation as it binds to PD-1 on activated T-cells and inhibits the interaction with its ligands, (PD-L1, PD-1L1) and (PD-L2, PD-1L2) [93]. A phase I/Ib (NCT03891953) study evaluating the efficacy of spartalizumab in combination with DKY709 (immunomodulatory agent) in patients with advances solid tumors including melanoma is ongoing [94]. A phase II PLATforM (NCT03484923) study evaluating the efficacy and safety of spartalizumab in combinations with LAG525 (monoclonal antibody targeting LAG-3), capmatinib (MET inhibitor), canakinumab (monoclonal antibody targeting IL-1β) and ribociclib (CDK4/6 inhibitor) is ongoing in previously treated unresectable or metastatic melanoma [95]. A phase III COMBI-i study (NCT02967692) comparing the combination of spartalizumab, dabrafenib and trametinib versus dabrafenib and trametinib in previously untreated patients with unresectable or metastatic BRAFV600 mutant melanoma is initiated [96]. 

## 5. Anti-CTLA-4

In addition to PD-1, another immune checkpoint inhibitor, cytotoxic T-lymphocyte antigen 4 (CTLA-4), is important in melanoma. It is found on the surface of regulatory T cells (Treg) and activated T cells [97]. CTLA-4 competes with CD28, another receptor expressed on the surface of T cells, to interact with its two ligands CD80 and CD86, collectively known as the B7 ligands. When CTLA-4 binds with the B7 ligands, commonly found on antigen presenting cells (APC), it results in an immunosuppressive response, which is the inhibition of T cell activation via transendocytosis of CD80 and CD86 from their surfaces [98,99]. Typically, T cell activation requires co-stimulation from the CD28-B7 ligand interaction and the TCR-MHC interaction [100]. However, CTLA-4 has a stronger affinity for the B7 ligands, making it a good immune checkpoint inhibitor that keeps the immune response from turning into an autoimmune one [97]. CTLA-4 is expressed on tumor cells, infiltrating Tregs, and exhausted, activated T cells [101]. Tumor cells, therefore, take advantage of this natural immunosuppressive system in order to prevent an immune response against them. This provides a therapeutic approach which involves anti-CTLA-4 therapy. There are currently three main anti-CTLA-4 antibodies under preclinical and clinical trials for the treatment of melanoma: Tremelimumab, Ipilimumab (Yervoy), and BCD-145. 

### 5.1. Tremelimumab/Ticilimumab/CP 675.206

This human monoclonal antibody against CTLA-4 was developed by AstraZeneca [102]. A phase I active, clinical trial (NCT02141542) is -evaluating tremelimumab in combination with MEDI3617 (human anti-angiopoietin 2 monoclonal antibody) for unresectable Stage III/IV melanoma patients [103]. Another phase I active, clinical trial (NCT01103635) is examining tremelimumab in combination with CP-870,893 CD40 agonist monoclonal antibody) for metastatic melanoma [104]. 

### 5.2. Ipilimumab/MDX010/BMS-734016

This human monoclonal antibody against CTLA-4 was developed by YERVOY Medarex/BMS. It was approved by the FDA in 2011 for the treatment of unresectable or metastatic melanoma [105]. There are current, active clinical trials devoted to assess the efficacy of ipilimumab in combination with other immunotherapies or targeted therapies for metastatic melanoma. A phase I clinical trial (NCT02115243) that assessed ipilimumab as a neoadjuvant followed by melphalan (chemotherapeutic) via isolated limb perfusion in patients with unresectable in-transit extremity melanoma is completed [106]. A phase Ib clinical trial (NCT02117362) evaluating ipilimumab in combination with GR-MD-02 (galnectin inhibitor) in metastatic melanoma patients has been completed [107]. A phase II clinical trial (NCT03153085) examining ipilimumab in combination with TBI-1401(HF10) in Japanese patients with Stage IIIb, IIIc, IV unresectable or metastatic malignant melanoma has been completed [108]. A phase II clinical trial (NCT01970527) looking at SBRT followed by Ipilimumab in patients with stage IV and recurrent melanoma has been completed [109]. 

### 5.3. BCD-145

This human monoclonal antibody against CTLA-4 is developed by BIOCAD [110]. A completed phase I clinical trial (NCT03472027) studied the efficacy of BCD-145 in unresectable/metastatic melanoma [111]. The combination of anti-PD-1/PD-L1 and anti-CTLA-4 are also being tested in the clinic for stage III/IV melanoma patients. A phase I clinical trial (NCT02935790) evaluating ipilimumab and nivolumab in combination with ACY-241 (selective HDAC inhibitor) is completed [112]. Current clinical trials, outcomes and adverse events investigating the efficacy of anti-CTLA-4, anti-PD-1/PD-L1 therapies and their combinations used to treat metastatic melanoma patients are listed in Table 1 [113,114,115,116,117,118,119,120,121,122,123,124,125,126,127,128,129,130,131,132,133,134,135,136,137,138,139,140,141,142,143,144,145,146,147,148,149,150,151,152,153,154,155,156,157,158,159,160].

## 6. AXL Inhibitors

The TAM family of receptor tyrosine kinases (RTKs) is comprised of **T**yro-3, **A**xl and **M**er (TAM). These TAMS regulate cell proliferation, survival, adhesion, migration, invasion and metastasis of neoplasms [161]. The AXL gene is located on chromosome 19q13.2; encoded by 20 exons. The protein structure consists of an extracellular domain consisting of a combination of two IgG like domains and two fibronectin type III repeats; a conserved intracellular kinase domain and a transmembrane domain [162,163]. Even though all three TAMS have transforming potential, the aberrant overexpression of Axl is associated with cancer progression, drug resistance and supports tumor immune escape in several cancers including melanoma [164,165,166,167,168,169,170,171,172]. In primary and acquired resistance in melanoma, Axl levels inversely correlate with levels of melanocyte lineage factor- Microphthalmia-associated transcription factor (MITF). The high Axl, low MITF drug resistance phenotype is found frequently among BRAF mutant melanoma cell lines. This is associated with a phenotype switch of cells form proliferative to an invasive phenotype and promotes metastasis [165,173,174]. The Axl inhibitors can be classified into 2 types. Type I encompasses inhibitors that compete with ATP and bind to the active conformation of the receptor, DGF-in (constitutes of the aspartate-phenylalanine-glycine (DFG) motif oriented towards the active site). Type II inhibitors interact with the DFG residues of the activation loop which open up an allosteric region, adopt an extended conformation and prefer binding to the inactive DFG-out conformation. [175]

### 6.1. BGB-324/Bemcentinib (Type I)

This highly selective orally bioavailable inhibitor was developed by BerGenBio [176]. Upregulation of Axl leads to drug resistance of BRAF directed therapies in the context of melanoma and also reduces response to PD-1 blockade. A Phase Ib/II trial is ongoing (NCT02872259) evaluating BGB324 in combination with dabrafenib/tramatenib or pembrolizumab in advanced non-resectable Stage IIIc/IV melanoma. The interim results for this study indicate that the combination was well tolerated at the recommended phase 2 dose of 200 mg daily of Bemcentinib. The common adverse events were diarrhea, fatigue, rash and pyrexia [177].

### 6.2. TP-0903 (Type I)

This oral Axl kinase inhibitor was developed by Tolero Pharmaceuticals, Inc. A first-in-human phase Ia/Ib trial (NCT02729298) evaluating TP-0903 in patients with advanced solid tumors encompassing BRAF mutated melanoma patients who haven’t responded to BRAF/MEK inhibitor combination or immunotherapy is currently recruiting patients [178,179].

### 6.3. Cabozantinib/XL184/BMS-907351 (Type II)

This inhibitor, developed by Exelixis [180], is an orally bioavailable small molecule inhibitor against various tyrosine kinases which include Axl, MET and VEGF. A phase II trial (NCT00940225) evaluating cabozantinib in patients with metastatic melanoma was discontinued as the study was underpowered to detect statistical significance [181]. A phase I/II trial (NCT03957551) evaluating the combination of cabozantinib and pembrolizumab as a front-line therapy has been initiated for patients with advanced metastatic melanoma [182].

### 6.4. LDC1267 (Type II)

This inhibitor preferentially inhibits Axl, Mer and Tyro3. In an in vivo model, it was observed that treatment with LDC1267 unleashes natural killer (NK) cells to target and kill tumor cells. Treatment with LDC1267 reduced the metastatic spreading of melanoma in an in vivo B16F10 melanoma model [183]. Further pre-clinical and clinical trials need to be initiated to test the efficacy of this drug in melanoma.

### 6.5. AXL-1047-MMAE

This is an antibody-drug conjugate (ADC) in which the Axl targeting human antibody is conjugated to monomethyl auristatin E (MMAE), a microtubule disrupting agent by a valine citrulline linker which is protease-cleavable. This ADC induces cytotoxicity in vitro and in vivo in melanoma models. Treatment with the ADC prevents the emergence of BRAF-inhibitor resistant clones and potentiates the efficacy of BRAF and MEK inhibitors and co-operatively targets the growth of resistant cells. This ADC along with BRAF and MEK inhibitors has shown efficacy in treatment naïve and MAPK pathway inhibitor resistant melanoma. A phase I/II trial (NCT02988817) evaluating enapotamab vendotin (HuMax-AXL-ADC) has been initiated in patients with solid tumors, including melanoma [184,185]

## 7. BRAF Inhibitors

BRAF inhibitors are small molecule inhibitors that selectively target mutant BRAF isoforms, preferentially V600E but also other isoforms such as V600K or V600D [186]. BRAF inhibitors are typically used in combination with inhibitors of MEK, the downstream target of BRAF, in order to delay the development of resistance to BRAF inhibitor monotherapy as in the current standard of care for late-stage BRAF^V600E^ melanoma, dabrafenib and trametinib. Vemurafenib/PLX4032/RG7204, a serine/threonine kinase inhibitor, was the first selective BRAF inhibitor that was approved by the FDA. It binds to the ATP-binding domain of BRAF mutants such as V600E, V600R and V600D [187]. 960 mg twice daily was established as the recommended phase 2 dose in the phase 1 (NCT00405587) dose escalation clinical trial [186]. The FDA approval was granted based on the Phase 3 trial (BRIM-3) results (NCT01006980) which exhibited improved overall survival and progression-free survival rate in patients with BRAFV600E mutant melanoma [188]. Dabrafenib, a type I-kinase inhibitor was the second BRAF inhibitor that was approved by the FDA. This reversible ATP-competitive inhibitor, inhibits BRAFV600E, V600D and V600K proteins [189]. The Phase 2 trial (BREAK-2; NCT01153763) trial established a dose of 150 mg twice daily which can either be used as a single agent or in combination with trametinib [190,191]. It was granted FDA approval on the basis of the outcomes of Phase 3 trial (NCT01227889) in which it exhibited improved progression-free survival vs. decarbazine [24].

### Encorafenib/LGX818

This molecule is an oral BRAF inhibitor selective for BRAF^V600E^ that was approved by the FDA in June 2018 for use in combination with the MEK inhibitor binimetinib (MEK162) in treating metastatic melanoma patients with the BRAF^V600E^ mutation [192]. A phase II trial (NCT02631447) to determine the optimal sequencing of BRAFi + MEKi (encorafenib + binimetinib) therapy and immunomodulatory antibody (ipilimumab + nivolumab) therapy in stage III-IV metastatic BRAF V600 melanoma is ongoing [193]. A phase II trial (NCT02159066) evaluating the use of third agent in encorafenib + binimetinib therapy in stage III-IV metastatic BRAF V600 melanoma [194].

## 8. MEK Inhibitors

MEK inhibitors are small molecule inhibitors targeting MEK1/2 proteins in the MAPK pathway. The addition of a MEK inhibitor in combination with a BRAF inhibitor delayed the development of resistance and decreased the toxicities associated with BRAF inhibitor monotherapy [195]. Trametinib/GSK1120212, a reversible, non-ATP-competitive inhibitor of MEK1/2 was the first MEK inhibitor approved by the FDA. The phase 1 study (NCT00687622) identified 2 mg daily dose of trametinib, which could be safety, administered to the patients [196]. The phase 3 COMBI-D trial (NCT01584648) provided evidence of combining dabrafenib and trametinib in patients with metastatic BRAFV600E/K mutant melanoma as compared to monotherapy with dabrefenib [197]. Cobimetinib/GDC-0973 is used in combination with vemurafenib and is approved for patients with BRAFV600E/K mutant metastatic melanoma. The FDA approval was granted based on the efficacy results of combination of vemurafenib and cobimetinib in Phase-3 co-BRM trial (NCT01689519) [198]. Binimetinib/MEK162 is used in combination with encorafenib and is used in patients harbouring BRAFV600E/K mutation.

### 8.1. KZ-001

This selective MEK1/2 inhibitor is a benzoxazole compound with high potency and exhibits anti-tumor activity in BRAF- and NRAS-mutant tumor cell lines. It presented a synergistic effect in in vitro and in vivo xenograft models when used in combination with docetaxel (microtubule-stabilizing chemotherapeutic agent) and vemurafenib [199].

### 8.2. E6201

This MEK1 inhibitor was developed by Eisai Inc. This ATP-competitive MEK inhibitor is a synthetic analog of a natural product f152A1 occuring from the fungus *Curvularia verruculosa* [200]. NCT00794781 was a Phase I trial, evaluating the efficacy and safety of E6201 in patients with BRAF mutant advanced melanoma was terminated early due to futility based on response data [201].

### 8.3. TAK-733

This selective, oral, potent, non-ATP competitive allosteric site MEK inhibitor was developed by Millenium Pharmaceuticals, Inc. It demonstrated anti-tumor effects in vitro in melanoma cell lines and in vivo in patients-derived xenograft models [202]. NCT00948467 was a Phase 1 dose escalation trial evaluating TAK-733 in advanced solid tumors including patients with advanced metastatic melanoma had manageable toxicity profile but had limited antitumor activity and based on this result the further investigations are not planned [203].

### 8.4. PD-0325901/Mirdametinib

This selective, potent, oral, noncompetitive MEK inhibitor was developed by Pfizer. It inhibited ERK phosphorylation in in vitro model. It inhibited growth of melanoma cell lines in vitro and in xenograft models. This molecule also inhibited angiogenesis by inhibiting VEGF production and induced apoptosis in in vitro models [204]. NCT00147550, a phase I/II clinical trial evaluating the efficacy of PD-0325901 in advanced melanoma has been terminated due to ocular, neurological and musculoskeletal toxicities at higher doses (> 15 mg twice a day) [205].

## 9. ERK Inhibitors

ERK plays a unique role in the MAPK/ERK pathway; it has more than 100 substrates, some of which are involved in MAPK/ERK activating/de-activating feedback loops, yet it has only one upstream effector, MEK1/2 [206]. Due to this role, ERK inhibitors may show promise as a method of overcoming the development of resistance and re-activation of BRAF and MEK in BRAF^V600E^ melanoma. While presently far behind BRAF and MEK inhibitors in development, there has been a recent increase in the development and evaluation of ERK inhibitors for treating BRAF^V600E^ melanoma.

### 9.1. Ulixertinib/BVD-523

Ulixertinib is a novel, selective ERK1/2 inhibitor developed by BioMed Valley Discoveries, that inhibits ERK1/2 in a reversible and competitive manner. Importantly, ulixertinib presented equivalent efficacy in BRAF mutant cells and BRAF + MEK double mutant cells, while the efficacy of BRAF and MEK inhibitors decreased in the double mutant line [207]. Currently, it has been designated for fast track status by the FDA in the treatment of metastatic BRAFV600E-mutant melanoma [208].

### 9.2. LY3214996

A selective ERK1/2 inhibitor developed by Eli Lilly currently in phase I clinical trials [209,210]. Further details of this pre-clinical characterization have not been made publicly available, and Phase I trials are on-going.

### 9.3. MK8353

An orally dosed, selective inhibitor of activated ERK1/2, and non-activated ERK2 developed by Merck & Co. currently recruiting for phase I trials [211,212]. A phase I clinical trial was initiated following these results in healthy volunteers (NCT01358331); however, the study was terminated after phase Ia MTD determination (400 mg orally once daily) for strategic reasons [213].

### 9.4. LTT462

An oral ERK inhibitor developed by Novartis. LTT-462 has completed a phase I clinical trial for use in advanced cancers, including melanoma. A phase I clinical trial (NCT02711345) evaluating the use of LTT462 in advanced melanoma and other advanced cancer has concluded; however, the results are not yet publicly available [214].

### 9.5. KO-947

A highly potent and selective ERK1/2 inhibitor developed by Kura Oncology [215].

### 9.6. GDC0994

An orally-dosed, selective ERK1/2 inhibitor developed by Genentech [216]. A phase Ia trial (NCT01875705) was conducted on MAPK-dysregulated cancers not including melanoma. The phase Ia trial found a safety profile consistent with MAPK inhibition with tolerable adverse events [217]. A following phase Ib trial (NCT02457793) investigating the use of GDC0994 in combination with cobimetinib (MEKi) in advanced cancers including advanced melanoma has completed; however, the comprehensive results have not yet been released [218].

### 9.7. SCH772984

This is a selective, ATP-competitive ERK inhibitor developed by Merck. It exhibits antitumor activity against BRAFV600E mutant and NRAS mutant melanoma. It blocks proliferation of melanoma cell lines in BRAF and MEK inhibitor resistant cell lines in vitro [219]. The synergistic combination of SCH772984 with Vemurafenib delayed the onset of acquired resistance in in vitro models [220]. Intermitent dosing with RAF inhibitor, MEK inhibitor and ERK inhibitor (SCH772984) inhibited tumor growth in low-level BRAF amplification patient derived xenograft model of melanoma [221].

Table 2 summarizes the clinical trials, outcomes and adverse effects of novel BRAF and ERK inhibitors that are under investigation in the clinic to treat metastatic melanoma patients [211,222,223,224].

## 10. ROS Activated Prodrugs

Redox homeostasis is essential for cell transcription, proliferation and survival. Failure of regulating redox homeostasis can cause DNA damage and cell apoptosis [225]. Cancer cells are known to express increased reactive oxygen species (ROS) such as superoxide, H_2_O_2_ and the hydroxyl radicals [226,227,228,229]. Evidence has shown a significant increase in ROS levels after B-RAF inhibition in melanoma cells [230]. ROS-activated prodrugs can be potentially utilized in combination with B-RAF inhibitors to target metastatic melanoma cells.

### 10.1. Protein Ribonuclease A (RNase A)/SN-38

These ROS activated drugs usually contain two separate functional domains- a ROS-accepting moiety, “Trigger”, and an “Effector”. In the presence of H_2_O_2_, the B-C bond within the aryl Byronic acid or esters will become oxidized, releasing the phenol group and activating the pro-drug. Protein ribonuclease A (RNase A) and SN-38 are being studied in the B16F10 murine melanoma cell line which mimics primary tumor growth [231,232]. SN-38 significantly decreases the proliferation of B16F10 murine melanoma cell line [231]. The enzymatic activity of RNase A will be reduced and cytotoxicity is improved when being activated via high ROS levels against skin melanoma cancer cells (B16F10) [232]. The success of these prodrugs in reducing tumor proliferation in murine melanoma cells gives the potential to study these drugs in human melanoma cell lines and potentially into clinical trials.

### 10.2. A100/RAC1

A100 is a quinone derivative. Quinones have substituents on the activated alkene, which are also called Michael acceptors. Cell damage and cytotoxicity occurs through the alkylation of DNA or cellular proteins. A100 sensitizes dabrafenib-resistant melanoma cells to BRAF protein kinase inhibitors [233]. A100, in the presence of high ROS levels, can self-cyclize into a bicyclic ring and cause DNA double strand breaks in cancer cells [234]. This compound and related ROS activated pro-drugs could be useful therapeutic agents where a BRAF inhibition has failed as the first line of treatment in melanoma patients harboring BRAF^V600E^ mutation.

## 11. Conclusions

Resistance to therapies continues to push the need to expand our understanding of melanoma treatment. This has led to exploring new treatments that utilizes combination therapies in order to achieve maximum anti-tumor efficacy over long durations of treatment avoiding resistance. Advances in the treatment of metastatic melanoma are on the rise with progress in targeted molecular therapy and immunotherapy. Targeted therapies are now expanding to include new BRAF and MEK inhibitors together and in combination with other therapies. Progress is being made in the field for targeting Axl and with ROS activated prodrugs. Immunotherapies are a new area of interest focusing on manipulation of checkpoint inhibition with durable clinical responses in patients. Melanoma has a strong molecular and genetic basis of pathogenicity, which allows for the development of personalized medicine. Selecting unique and individual treatments for melanoma patient makes it more likely to achieve high success rates in the clinic. Continual research and clinical trials are ongoing to further elucidate and expand knowledge on mechanisms of resistance and novel treatment strategies such as immunotherapies, new small molecule inhibitors and ROS-activated prodrugs to provide effective care to patients with metastatic melanoma.

## Figures and Tables

**Figure 1 cancers-12-00482-f001:**
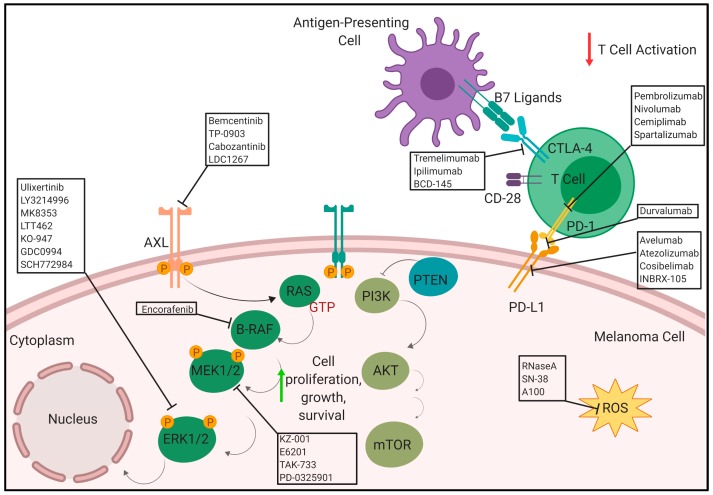
Novel therapies in pre-clinical and clinical phases (Anti PD-1/anti-PD-L1, anti CTLA-4, AXL inhibitors. BRAF inhibitors. ERK inhibitors and ROS activated prodrugs) to treat patients with metastatic melanoma (created with BioRender; www.biorender.com).

**Table 1 cancers-12-00482-t001:** Summary of clinical trials, outcomes and adverse events associated with anti-PD-1/PD-L1, anti-CTLA-4 and their combination in metastatic melanoma patients.

Treatment	Status	Sponsor	Phase and NCT	Clinical Outcomes	Adverse Events
Anti-PD-1/PD-L1					
Pembrolizumab in Japenese patients with advanced melanoma (KEYNOTE-041) [113]	Completed	Merck Sharp & Dohme Corp	Ib, NCT02180061	Pembrolizumab has promising anti-tumor activity and an acceptable safety profile in patients with cutaneous melanoma (*n* = 29). As per central review, the median overall survival (OS) and median duration of response was not reached. The 1 year OS was 82.7%. The median profession-free survival (PFS) and 6 months PFS was 4.2 months (95% CI: 2.8–7 months) and 41.4% respectively. The overall response, complete response (CR) and partial response (PR) were 24.1% (95% CI: 10.3–43.55), 6.9% (95% CI: 0.8–22.8%) and 17.2% (95% CI: 5.8–35.8%) respectively.	Pruritus, anemia, maculopapular rash, malaise, and hypothyroidism
Study of pembrolizumab (MK-3475) versus chemotherapy in patients with advanced melanoma KEYNOTE-002) [114]	Completed	Merck Sharp & Dohme Corp.	II, NCT01704287	The progression-free survival was improved in patients assigned to pembrolizumab 2 mg/kg (HR 0.57, 95% CI 0.45–0.73; *p* < 0.0001) and those assigned to pembrolizumab 10 mg/kg (0.50, 0.39–0.64; *p* < 0.0001) compared with those assigned to chemotherapy. 6-month progression-free survival was 34% (95% CI 27–41) in the pembrolizumab 2 mg/kg group, 38% (31–45) in the 10 mg/kg group, and 16% (10–22) in the chemotherapy group.	Fatigue, generalised oedema, myalgia, hypopituitarism, colitis, diarrhoea, anemia decreased appetite, hyponatremia, pneumonitis, neutropenia and leucopenia.
Pembrolizumab versus ipilimumab in advanced melanoma (KEYNOTE-006) [115]	Completed	Merck Sharp & Dohme Corp.	III, NCT01866319	Pemrolizumab showed superiority over ipilimumab at 5 year follow up time. 834 patients were stratified into three groups: (i) Pembrolizumab (10 mg/kg i.v. every 2 weeks), (ii) Pembrolizumab (10 mg/kg i.v. every 3 weeks) and Ipilimumab (3 mg/kg i.v. every 3 weeks). The median follow up was 57.5 months (IQR: 56.7–59.2 months). Combined Pembrolizumab groups: The median OS and PFS were 32.7 months (95% CI: 24.5–41.6 months) and 8.4 months (95% CI: 6.6–11.3 months) respectively. The ORR was 42% (95% CI: 38.1–46.5%). Ipilimumab groups: The median OS and PFS were 15.9 months (95% CI:13.3–22 months; HR 0.73; 95% CI for HR- 0.61–0.88; *p* = 0.00049) and 3.4 months (95% CI: 2.9–4.2 months; HR 0.57; 95% CI for HR: 0.48–0.67; *p* < 0.0001) respectively. The ORR was 17% (95% CI: 12.4–21.4%).	Fatigue, colitis, diarrhea, asthenia, arthralgia, rash, pruritus, vitiligo.
Durvalumab in combination with Dabrafenib and Trametinib in patients with advanced melanoma [116]	Completed	MedImmune LLC	I. NCT02027961	Durvalumab in combination with dabrafeib and trametinib had manageable safety profile. No maximum tolerated dose was identified (*n* = 50) and durvalumab 10 mg/kg was selected for further studies.	Pyrexia, fatigue, diarrhea, rash, vomiting and other drug related toxicities
Nivolumab in metastatic melanoma patients [117]	Active, not recruiting	BMS in collaboration with Ono Pharmaceutical Co. Ltd.	I, NCT00730639	Treatment with nivolumab is associated with long-term survival in patients with melanoma (*n* = 72). The median duration to response and objective response rate (ORR) was 22.9 months (95% CI: 19.7–31.8 months) and 31.8% respectively. The median, estimated 3 years and 5 year overall survival rates were 20.3 months (95% CI: 12.5–37.9 months), 42.3% (95% CI: 32.7–51.6%) and 34.2% (95% CI: 25.2–43.4%) respectively. Patients who had an ORR had significantly higher mean baseline absolute lymphocytes count (1480 cells/uL) as compared to patients without response (1300 cells/uL; *p* = 0.4).	Anemia, hypothryoididm, gastrointestinal disorder, general disorder, muscular disorder, nasopharyngitis, decreased apatite, nervous and respiratory problems, vascular and skin disorder
Atezolizumab in combination with vemurafenib alone or in combination with cobimetinib [118]	Active, not recruiting	Genentech, Inc.	Ib, NCT01656642	The triple combination was safe, tolerable and had a promising anti-tumor activity. Atezolimumab + Vemurafenib (*n* = 17): The best objective response rate and complete response rate was 76.5% (95% CI: 50.1–93.2%) and 17.6% respectively. All the patients demonstrated a reduction in the sum of the longest diameter of the target lesion. The median duration of response, PFS and OS was 10.6 months (95% CI: 9.1–37.6 months), 10.9 months (95% CI: 5.7–22 months) and 46.2 months (95% CI: 24.1-not reached) respectively. Estimated OS rates for 1 year were 82%. Atezolimumab + Vemurafenib + Cobimetinib (*n* = 39): The best objective response rate and complete response rate was 71.8% (95% CI: 55.1–85%) and 20.5% respectively. All the patients demonstrated a reduction in the sum of the longest diameter of the target lesion. The median duration of response and PFS was 17.4 months (95% CI: 10.6–25.3 months), 12.9 months (95% CI: 8.7–21.4 months) respectively. The median OS was not reached. Estimated OS rates for 1 year were 83%. Treatment with vemurafenib alone or in combination with cobimetinib exhibited an increase in the proliferating CD4+ T-helper cells and addition of atezolizumab led to further escalation in these cells. CD8+ cytotoxic T cells were augmented on addition of atezolizumab.	Increase in AST, ALT and blood bilirubin, hyponatremia, blood alkaline phosphatase, rash, diarrhea, vomiting.
Anti-CTLA-4					
Tremelimumab in patients with advanced refractory and/or relapsed melanoma [119]	Completed	AstraZeneca	II, NCT00254579	Tremelimumab showed a durable response in these patients (*n* = 241). The ORR was 6.6% and the duration of response was 8.9–29.8 months. The median OS and clinical benefit rate was 10.1 months (95% CI: 7.9–11.7 months) and 21% respectively.. The survival rate at 1 and 2 years was 40% (95% CI: 34–46%) and 22% (95% CI: 17–27%) respectively. Median PFS and 6 months PFS was 2.8 months (95% CI: 2.7–2.8 months) and 15% respectively. As per Response Evaluation Criteria in Solid Tumors (RECIST) criteria, 3.3% of the patients achieved response at the target lesion.	Diarrhea, pruritus, rash, nausea, fatigue, vomiting and colitis.
Ipilimumab alone or in combination with dacarbazine, paclitaxel and carboplatin [120]	Completed	BMS in collaboration with Medarex	I, NCT00796991	Ipilimumab could be combined safely with these chemotherapies with no major pharmacokinetic/pharmacodynamic interactions being observed in these patients. The combinations exhibited a good anti-tumor activity. Ipilumumab alone (*n* = 20): Estimated geometric mean for Area Under the Curve (AUC) (0-infinity) and maximum serum concentration (C_max_) for Ipilimumab metabolite (AIC) in presence of ipilimumab was changed by 0.970 (90% CI: 0.891–1.056) and 1.058 (90% CI: 0.974–1.150) fold respectively. Based on World Health Organization (WHO) and immune-related criteria ORR were 29.4% and 33.3% respectively and disease control rates b were 59.2% and 73.3% respectively. Ipilimumab + Decarbazine (*n* = 19): Estimated geometric mean for AUC (0-infinity) and C_max_ for dacarbazine in presence of ipilimumab was changed by 0.912 (90% CI: 0.757–1.099) and 1.027 (90% CI: 0.848–1.243) folds respectively. Based on WHO and immune-related criteria ORR were 27.8% and 33.3% respectively, and disease control rates were 55.6% and 61.1% respectively. Ipilimumab + paclitaxel + carboplatin (*n* = 20): Estimated geometric mean for AUC (0-infinity) and C_max_ for carboplatin/paclitaxel in presence of ipilimumab was changed by 0.970 (90% CI: 0.891–1.056) and 1.058(90% CI: 0.974–1.150) folds respectively. ORR based on WHO and immune related criteria were 11.1% and 27.8% respectively. Disease control rate based on WHO and immune related criteria were 44.4% and 55.6% respectively. There was a significant increase in the mean relative frequency and counts of HLA-DR+ CD4+ and CD8+ T cells after treatment initiation in all the three groups.	Rash, fatigue, diarrhea, pruritus, nausea, increase in ALT and AST, decreased neutrophil count.
Ipilimumab in combination with imatinib mesylate in patients with advanced malignancies [121]	Completed	M.D. Anderson Cancer Center in collaboration with NCI	I, NCT01738139	The combination was well tolerated and safe and the MTD and recommended phase 2 dose for intravenous ipilimumab was 3 mg/kg every 3 weeks and imatinib mesylate at 400 mg orally twice daily. Twenty six patients were enrolled in dose escalation cohort Expression of ICOS and OX40 was increased on the CD4 + T cells upon ipilimumab treatment.	Fatigue, nausea, vomiting, edema, anemia, diarrhea, rash, fever.
Ipilimumab and high dose IFN-α2B as a neoadjuvant combination for locally/regionally advanced/recurrent melanoma [122]	Completed	Diwakar Davar, University of Pittsburgh	I, NCT01608594	The combination was well tolerated and exhibited promising durable clinical response rates. 30 patients were enrolled. The median follow-up was 32 months and the pathologic complete response rate was 32% (95% CI: 18–51%). The radiologic response rate was 36% (95% CI: 21–54%). The median PFS was not reached and the probability of PFS at 12 and 6 months was 0.79 (95%CI: 0.65–0.95) and 0.86 (95% CI: 0.74–1) respectively. The probability of OS at 2 years and 1 year was 0.89 (95% CI: 0.79–1) and 0.93 (95% Ci: 0.84–1) respectively. The peripheral blood mononuclear cell was significantly lower at 12 weeks (*p* = 0.025). The tumor-infiltrating lymphocyte (TIL) was significantly higher in primary melanoma tumors for patients with pathologic complete response (*p* = 0.033). There was an increase in the number of tumor associated clones following the neoadjuvant treatment and it showed a strong correlation with TIL fraction (ρ = 0.7299; *p* = 0.0003) and TIL clone diversity (ρ= 0.882; *p* = 2.7–7). The increase in the tumor T-cell clonality in the primary tumor and a further increase in the clonality after neoadjuvant therapy was statistically significant with relapse-free survival. (*p* = 0.048 for tumor clonality and *p* = 0.018 for post treatment metastatic clonality).	Fever, fatigue, creatinine increase, skin, GI, hepatic, endocrine and hematologic disorders.
Ipilimumab in combination with PEG-Intrerferon (IFN) α2B on for Stage IIb/c/IV unresected melanoma [123]	Completed	H. Lee Moffitt Cancer Center and Research Institute in collaboration with Merck Sharp & Dohme Corp.	Ib, NCT01496807	The maximum tolerated dose established was 3 mg/kg of Ipilimumab and 2 ug/kg/week of peginterferon alfa-2 b was efficacious and well tolerated. 30 patients were enrolled. Immune related response criteria: 3.33% and 36.67% of the subjects achieved a CR and PR respectively. The overall response rate was 40%. The median follow up time was 35.8 months (Range: 19.7–50.2 months). The median PFS was 5.9 months and median OS was not reached. At 40.3 months, 16.7% of the patients had a prolonged PFS without any need for further therapy. 85.6% of the subjects had an objective response. Here was a significant correlation between autoimmune vitiligo and objective response (*p* = 0.009).	Anemia, dry eye, GI disorders, chills, fatigue, fever, increase in blood enzymes, anorexia, muskoskeletal and connective tissue disorders, nervous system disorders. Cough, dyspnea, depression, skin and subcutaneous tissue disorders
Intratumoral injection of Ipilimumab and IL-2 for unresectable stage III/IV melanoma [124,125]	Completed	University of Utah	I, NCT01672450	The combination was well tolerated and generated an enhanced systemic immune response at injected and non-injected lesions in these patients. No dose limiting toxicities were observed in the 12 enrolled patients. Immune-related response criteria: Clinical benefit rate was 50% (95% CI: 19–81%). The PR and overall ORR was 30% and 40% (95% CI: 10–70%) respectively. 67% of the subjects (95% CI: 40–94%) had local response on the injected lesion which was assessed by pathology and/or measurement. 88.9% of the patients (95% CI: 68–100%) had an abscopal effect observed at distant and locoregional sites. Based on imaging and/or pathology, 40% of the patients (95% CI: 10–70%) demonstrated an objective response. 60% of the subjects had an increase in the frequency of CD8+ T cells expressing Tbet (fold increase: 1.75; 95% CI: 1.14–2.36) and IFNgamma (fold increase: 2.02; 95% CI: 1.31–2.73). 40% (fold increase: 1.83; 95% CI: 1.61–2.05) and 50% (fold increase: 1.50; 95% CI: 1.32–1.68) of the patients had an increase in the CD8+ T cells expressing granzyme-B and perforins.	Chills, fatigue, flu like symptoms, pain and ulceration at site of injection, rash, soft tissue infection. No grade 4/5 toxicities were observed.
Ipilimumab in combination with Stereotactic Radiosurgery (SRS) or Whole Brain Radiation Therapy (WBRT) [126]	Completed	Sidney Kimmel Cancer Center at Thomas Jefferson University in collaboration with BMS	I, NCT01703507	The combination of ipilimumab (3 mg/kg; *n* = 7 or 10 mg/kg; *n* = 9) and SRS was safe without any dose limiting toxicities. Arm A (WBTR; *n* = 5): The median follow-up time was 8 months (Range: 3.5–24.1 months). Median time to intracranial progression, PFS and OS were 2.53 months (Range 0.3–18 months), 2.5 months and 8 months respectively. Arm B (SRS; *n* = 11): the median follow-up time was 10.5 months (Range: 1.8–36.8 months). Median time to intracranial progression and PFS was 2.45 months (Range: 1–37 months) 2.1 months respectively. The median OS was not reached. Immune-related PR was 7%.	Neurotoxic effects, headache, GI toxicity, vomiting, subclinical intracranial hemorrhage, increase in ALT, dizziness, tinnitus. No grade 4/5 and radionecrosis were observed.
Ipilimumab in subjects with previously treated unresectable stage III/IV melanoma [127]	Completed	BMS in collaboration with Medarex	II NCT00289640	The three fixed doses used in this study were ipilimumab 10 mg/kg, 3 mg/kg or 0.3 mg/kg administered every 3 weeks for four cycles (induction phase) followed by maintenance therapy administered every 3 months. Ipilimumab had a dose-dependent efficacy and safety in these subjects and 10 mg/kg dose was well tolerated with manageable safety. 10 mg/kg (*n* = 73): The median follow-up was 10.7 months (Interquartile range {IQR}: 3.6–20.4 months). Best overall response rate was 11.1% (95% CI: 4.9–20.7). This dose had greater increase in absolute lymphocyte count and serum ipilimumab concentrations compared to the other doses at 4 weeks. 3 mg/kg (*n* = 72): The median follow-up was 8.3 months (IQR: 4–20.4 months). Best overall response was 4.2% (95% CI: 0.9–11.7). 0.3 mg/kg (*n* = 32): The median follow-up was 8.3 months (IQR: 3.5–15.3 months). Best overall response was 0% (95% CI: 0–4.9). The best overall response was significant for 10 mg/kg group (*p* = 0.0015). Median OS for 10 mg/kg vs 3 mg/kg had an HR of 0.875 (95% CI: 0.593–1.291). Median OS for 10 mg/kg vs 0.3 mg/kg had a hazard ratio (HR) of 0.77 (95% CI: 0.525–1.130).	Immune-related grade 3–4 events were GI related, skin related, nausea, vomiting, pruritus, rash, endocrine. Some immune related grade 5 adverse events were observed.
Ipilimumab in combination with Fotemustine for unresectable locally advanced or metastatic malignant melanoma with or without brain metastasis (NIMIT-M1) [128]	Completed	Italian Network for Tumor Biotherapy in collaboration with BMS	II, NCT01654692	The combination was efficacious and fotemustine did not impair the activity of Ipilimumab. The median follow up time was 39.9 months. The median duration of response, 3 and 2 year duration of response rates were 30.3 months (95% CI: 15.5–46.5 months), 49.2% (95% CI: 27.4–71%) and 55.4% (95% CI: 34.7–76.1%). Whole Study Population (*n* = 86): The median OS was 12.9 months (95% CI: 7.1–18.7 months). The 3 and 2 year survival rates were 28.5% and 33.4% respectively. The PFS and median brain PFS were 4.5 months (95% CI: 3.1–5.9 months) and 8.3 months (95% CI: 4.7–11.8 months) respectively. For patients with brain metastasis: The median OS, 3 and 2 years survival rates were 12.7 months (95% CI: 2.7–22.7 months), 27.8% and 38.9% respectively. The PFS and median brain PFS were 3.4 months (95% CI: 2.3–4.5 months) and 3 months (95% CI: 2.9–3.1 months). The median OS was not significant for patients with BRAF WT and V600E mutation. Subjects with improved OS had increased levels of circulating CD3 ^+^ CD4^+^ICOS^+^CD45RO^+^T cells as opposed to CD3^+^CD4^+^ICOS^+^CD45RA^+^ T cells. The expansion of memory T cells over naïve T-cells shows that the combination favors an increase in T-cell antigen-primed populations.	Rash, pruritus
Ipilimumab in advanced melanoma patients with preexisting humoral response to NY-ESO-1 [129]	Completed	National Center for Tumor Diseases, Heidelberg in collaboration with University Hospital Heidelberg	II, NCT01216696	Ipilimumab demonstrated a higher clinical efficacy in patients with NY-ESO-1, maybe used as a surrogate for preexisting immune response to tumor antigens. 25 patients were enrolled in the study. The median duration of treatment was 64 days (Range: 1–352 days). Immune-related response criteria: the disease control rate was 52% (90% CI: 34.1–69.5%). 36% of the subjects had a PR. The PFS was 7.8 months (95%CI: 2.6-nr moths). No significant association was observed between best response and the amount of NY-ESO-1-specific T-cells. RECIST criteria: 24% of the subjects had a PR. The PFS was 2.9 months (95%CI: 2.5–8.1 months). The median OS was 22.7 months (95%CI: 9.5-nr months). The 1 year survival rate was 66.8% (95%CI: 0.44–0.82). The best overall response (BOR) had a statistically significant association with the OS (*p* = 0.0031). For a small subset of patients there was a statistically significant association between PFS and NY-ESO-1-specific CD3+ T-cells (HR: 1.039; *p* = 0.0478).	Endocrine, GR, hepatobiliary, musculoskeletal and connective tissue disorders, headache, skin and subcutaneous tissue disorders
Vemurafenib followed by ipilimumab in V600 BRAF mutant, untreated metastatic melanoma patients [130]	Completed	BMS	II, NCT01673854	The study was divided into two parts: one where patients received vemurafenib followed by ipilimumab and the other where subjects who progressed after ipilimumab received vemurafenib. The sequential treatment was efficacious and had a manageable safety profile. The use of targeted therapy followed by immune modulation therapy has helped to understand the optimum regimen of these therapies. VEM1-IPI: 46 patients were treated with vemurafenib followed by 46 patients on ipilimumab induction and eight patients on ipilimumab maintenance. The median duration of response and follow-up was 23.1 months (95%CI: 5.03-not reached). The BOR was 32.6%. The median PFS and OS was 4.5 months (95%CI: 4.17–6.57 months) and 18.5 months (95%CI: 11.96-not evaluated). VEM2: 19 patients progressed on pilimumab were treated with vemurafenib. The median follow-up and overall survival was 15.3 months and 18.5 months (95% CI: 11.96-not evaluated) respectively. The median PFS was 4.5 months (95% CI: 4.17–6.57 months). The BOR rate was 36.8%. 4.3% of patients had a CR and 28.3% had a PR.	Rash, erythema, pruritus, GI toxicities, hetaobiliary toxicities, nausea, vomiting.
Ipilimumab in combination with HF-10 (replication –competent HSV-1 oncolytic virus) in stage IIIb/c/IV unresectable or metastatic malignant melanoma [131]	Completed	Takara Bio Inc. in collaboration with Theradex	II, NCT02272855	The combination was well tolerated, beneficial and elicited anti-tumor activity. The combination induced an immune-cell infiltration in the TME. 46 patienrs were enrolled. The best overall response rate was at 24 weeks. Immune-related response criteria: 18% and 23% of the patients had a CR and PR respectively. The median PFS and OS were 19 months and 26 months respectively. There was increase om the total tumor infiltrating CD8+ T-cells and lymphocytes along with a decrease in the CD4+ T-cells.	Treatment related grade 3/4 events. Majority of the AEs were due to Ipilimumab which are immune related events.
Ipilimumab in combination with standard melphalan and dactinomycin to isolate limb infusion (ILI) for advanced unresectable melanoma of the extremity [132]	Completed	Memorial Sloan Kettering Cancer Center in collaboration with BMS	II, NCT01323517	The combination was safe and efficacious. 18 patients were enrolled. The median follow-up time was 18 months. At 3 months timepoint, 89% of the patients had a limb response from which 65% and 24% had a CR and PR, respectively. At 18 months, the median OS was 78%. The PFS at one year was 57%. The levels of eosinophils and ALC were elevated in all the subjects.	Limb toxicity, colitis, hypophysitis, rash.
Ipilimumab in Japanese patients with unresectable or metastatic melanoma [133]	Completed	BMS	II, NCT01990859	Ipilimumab was well tolerated and demonstrated an anti-tumor response in these patients (*n* = 20). The median OS and PFS was 8.71 months (95% CI: 3.71-nr months) and 2.74 months (95% CI: 1.25–2.83 months) respectively. The disease control rate and best overall response rates were 20% (95% CI: 5.7–43.7) and 10% (95% CI: 1.2–31.7) respectively. 10% of the subjects had a CR.	Rash, pruritus, pyrexia, GI disorder, increase in AST and ALT, skin, liver and endocrine related immune events. No grade 4 drug related adverse events were observed.
Ipilimumab in combination with temozolamide in metastatic melanoma patients [134,135]	Completed	M.D. Anderson Cancer Center in collaboration with BMS	II, NCT01119508	The combination exhibited an enhanced antitumor activity. 64 patients were enrolled. The median duration of response and median follow-up was 35 months (Range: 2–57 months) and 20 months (Range: 2–60 months). 15.6% of the subjects had a CR and PR. The median PFS and OS was 5 months and 24.5 months respectively. 6 months PFS was 45%. The PFS for patients with bone metastasis was significantly decreased (*p* = 0.014) but not for subjects with liver metastasis. 21% and 7% of the subjects with liver metastasis had a CR and PR respectively while no ORR was observed in subjects with bone metastasis	Pruritus, skin rash, nausea, constipation, diarrhea, colitis, increase in ALT and AST, hematologic toxicities. No drug related grade 5 toxicities were observed.
Ipilimumab as monotherapy for previously treated unresectable stage III/IV melanoma [136,137]	Completed	BMS in collaboration with Medarex	II, NCT00289627	Ipilimumab demonstrated good clinical activity in these patients which consists of subjects who did not respond to prior therapy. 155 patients were enrolled in the study. The median follow-up and OS was 10 months (Range 0.32–33.1 months) and 10.2 months (95% CI: 7.6–16.3%). The best overall response rate and disease control rate was 5.8% (95% CI: 2.7–10.7%) and 27% (95% CI: 20–35%) respectively. 5.8% of the subjects had a PR and no CR was observed. The 1 year, 18 months and 2 year survival rates were 47.2% (95% CI: 39.5–55.1%), 39.4% (95% CI: 31.7–47.2%) and 32.8% (95% CI: 25.4–40.5%), respectively.	Immune related grade 3 and 4 events - skin and GI tract, liver and endocrine. No grade 5 immune related adverse events were observed.
Ipilimumab in combination with autologous TriMix –DC therapeutic vaccine for previously treated unresectable stage III/IV melanoma [138,139]	Completed	Bart Neyns, Universitair Ziekenhuis Brussel in collaboration with Vrije Universiteit Brussel	II, NCT01302496	The combination was well tolerated and demonstrated a durable anti-tumor response. 39 patients were enrolled in the trial. The median follow-up time, estimated median PFS and OS were 36 months (Range: 22–43 months), 27 weeks (95% CI: 9–44 weeks) and 59 weeks (Range: 72–185 weeks) respectively. PFS and OS rates at year one- 33% (95% CI: 18–48%), 59% (95% CI: 43–74%); year two- 22%(95% CI: 9–36%), 38%(95% CI: 23–53%) and year three- 18% (95% CI: 5–31%), 34% (95% CI: 19–50%). Immune-related response criteria: 6 months disease control rate was 51% (95% CI: 36–67%). The ORR was 38%. 20% and 18% of the subjects had a CR and PR respectively. The outcome was poor in patients with brain metastasis. There was a significant increase in the eosinophils, peripheral blood lymphocytes and monocytes after from the baseline treatment with the combination. A significant increase in the CD8+ (*p* < 0.001), CD4+ (*p* < 0.001), HLA-DR+ activation marker on CD4+ cells (*p* < 0.001), CD3+ (*p* < 0.001), ratio of CD4+ to CD8+ (*p* = 0.003) and Tregs (*p* = 0.016) was observed after the combination treatment.	Dermal injection site reactions, post-infusion chills, flu-like symptoms. Immune related adverse events- dermatitis, colitis, diarrhea, hypophysitis, neumonitis, lymphadenopathy. Grade 3/4 immune related adverse events were observed with no grade 5 events.
Ipilimumab in combination with WBRT for melanoma patients and brain metastases (GEM STUDY, GRAY-B) [140,141]	Completed	Grupo Español Multidisciplinar de Melanoma in collaboration with BMS	II, NCT02115139	The combination was well tolerated and safe. Fifty eight patients were enrolled in the study. The overall survival at 1 year was 31.8% (95% CI: 18.8–44.8%). The median OS and PFS was 5.8 months (95% CI: 3.6–5.9 months) and 4.8 months (95% CI: 2.2–3.4 months).	WBRT related- vomiting and headache. Immune therapy related- diarrhea, intestinal perforation, increase in AST and ALT, headache.
Ipilimumab in combination with SRS for subjects with brain metastasis [142]	Completed	University Hospital, Lille in collaboration with BMS	II, NCT02662725	High dose of ipilimumab and SRS was effective with a manageable safety profile. 57 patients were enrolled in the study. Median survival time for reference population vs study population was 5.6 months vs 13.2 months (HR: 0.29, 95% CI: 0.19–0.39; *p* < 0.0001) with 49% disease control rate.	Colitis, hepatitis, hypophiisitis, headache. One subject showed radionecrosis.
Ipilumumab alone or in combination with decarbazine in previously untreated metastatic melanoma patients [143]	Completed	BMS	II, NCT00050102	The combination was well tolerated and a durable, clinically meaningful responses were observed. Decarbazine did not affect the PK of ipilimumab. Ipilimumab alone (*n* = 37): The median follow up time and median OS were 16.4 months and 11.4 months (95% CI: 6.1–15.6 months) respectively. The objective response rate and disease control rate were 5.4% (95% CI: 0.7–18.2%) and 21.6% (95% CI: 9.8–38.2%), respectively. 5.4% of the subjects had a PR. The survival rates for 1 year, 2 year and 36 months were 45%, 21% and 9%, respectively. Combination (*n* = 35): The median follow up time and the OS were 20.9 months and 14.3 months (95% CI: 10.2–18.8 months) respectively. The objective response rate and disease control rate was 14.3% (95% CI: 4.8–30.3%) and 37.1% (95% CI: 21.5–55.1%) respectively. 5.7% and 8.6% of the patients had a CR and PR for more than 24 weeks respectively. The survival rates for 1 year, 2 year and 36 months were 62%, 24% and 20% respectively. CD4+ and CD8+ expressing HLA-DR T cells were increased in both the groups.	Colitis, muscle weakness, anemia, tachycardia, abdominal pain. Fatigue, dehydration, increased ALT/AST, rash, pruritus, vasculitis.
Ipilimumab in combination with HF10 for unresectable Stage IIIb/c/IV or metastatic malignant melanoma [144]	Completed	Takara Bio Inc. in collaboration with Theradex	II, NCT02272855	The combination was well tolerated with positive antitumor activity and there were no dose limiting toxicities. 46 patients were enrolled. The median PFS and OS was 19 months and 21.8 months respectively. Immune-related response criteria: The best overall response rate and disease stability rate was 41% and 68%, respectively. 16% and 25% of the patients had CR and PR, respectively.	Embolism, lymphedema, diarrhea, hypoglycemia, groin pain, immune related events.
Decarbazine alone or in combination with ipilimumab for unresectable stage III/IV melanoma [145]	Completed	BMS in collaboration with Medarex	II, NCT00324155	The combination of ipilimumab and decarbazine was well tolerated and had a long-term durable overall survival. Ipilimumab and decarbazine group (*n* = 250): The median survival follow-up time and OS was 11 months (Range: 0.4–71.9 months) and 11.2 months (95% CI: 9.5–13.8 months) respectively. At 5 years, 18.2% of the patients were alive which was significantly higher than that in the other group (*p* = 0.002). 7.5% and 42.5% of the patients had a CR and PR respectively. No median OS was reached for the responders while that for the non-responders was was 14.3 months (95% CI: 11.4–16.9 months; HR: 0.28, 95% CI: 0.16–0.47). Decarbazine and placebo group (*n* = 252): The median survival follow-up time and OS was 8.9 months (Range: 0.1–73.2 months) and 9.1 months (95% CI: 7.8–10.5 months; HR: 0.69; 95% CI: 0.57–0.84) respectively. At 5 years, 8.8% of the patients were alive. 35% of the patients had a PR and no CR was achieved. The median OS for non responders and responders was 12.3 months (95% CI: 10.9–15.4 months) and 20.2 months (95% CI: 14.6–45.3 months; HR: 0.51, 95% CI: 0.32–0.84), respectively.	Rash, pruritus, vitiligo, GI, liver and endocrine related events. Grade 3 to 4 immune related adverse events were observed in skin with no grade 5 events.
Ipilimumab antibody (MDX-010) alone or in combination with melanoma peptide vaccine (MDX-1379; gp100) for previously untreated unresectble stage III/IV melanoma [146]	Completed	BMS	III, NCT00094653	Overall ipilimumab resulted in survival of 20% of the patients for more than 2 years. 45% of the patients who survived for more than 2 years survived for more than 3 years. Ipilimumab + placebo (*n* = 137): 25% of the patients survived for more than 2 years and 3 years. The disease control rate for on-study response and for patients surviving more than 2 years was 28.5% (1.5% CR and 9.5% PR) and 83.3% (8.3% CR and 41.7% PR) respectively. Gp100 vaccine alone (*n* = 136): 17% and 10% of the patients survived for more than 2 year and 3 years respectively. The disease control rate for on-study response and for patients surviving more than 2 years was 11% (1.5% PR) and 43.8%, respectively. Combination (*n* = 403): 19% and 15% of the patients survived for more than 2 years and 3 years respectively. The disease control rate for on-study response and for patients surviving more than 2 years was 20.1% (0.2% CR and 5.5% PR) and 66.7% (1.9% CR and 22.2% PR), respectively.	Immune related adverse events- colitis, vitiligo, diarrhea, hypogonadism, proctitis, dermatologic, GI, endocrine related events, increase ALT. No grade 4/5 immune related adverse effects were observed.
Ipilimumab doses- 3 mg/kg vs 10 mg/kg for previously treated or untreated unresectable or metastatic melanoma [147]	Completed	BMS	III, NCT01515189	Both doses were well tolerated, with 10 mg/kg dose having more treatment-related events. 10 mg/kg group (*n* = 364): The median follow-up time, OS and PFS were 14.5 months (IQR: 4.6–42.3 months), 15.7 months (95% CI: 11.6–17.8 months) and 2.8 months (95% CI: 2.8–3 months) respectively. 2% and 13% of the patients had a CR and PR respectively. The 1 year, 2 year and 3 year overall survival was 54.3% (95% CI: 49–59.3%), 38.5% (95% CI: 33.4–43.5%) and 31.2% (95% CI: 26.4–36%), respectively. 3 mg/kg group (*n* = 362): The median follow-up time, OS and PFS was 11.2 months (IQR: 4.9–29.4 months), 11.5 months (95% CI: 9.9–13.3 months) and 2.8 months (95% CI: 2.8–2.8 months), respectively. 2% and 10% of the patients had a CR and PR respectively. The 1 year, 2 year and 3 year overall survival was 47.6% (95% CI: 42.4–52.7%), 31% (95% CI: 26.2–35.8%) and 23.2% (95% CI: 18.9–27.7%), respectively. HR between both the groups for median overall survival was 0.84 (95% CI: 0.7–0.99; *p* = 0.04). EORTC QLQ-C30 global health status were significantly declined in both the groups form the baseline.	Headache, diarrhea, colitis, increase in ALT, hypophysitis. No grade 5 toxicities were observed.
Ipilimumab alone or in combination with talimogene laherparepvec (T-VEC) in patients with previously untreated unresected, Stage IIIb-IV melanoma [148]	Active, not recruiting	Amgen	Ib/II, NCT01740297	The combination was well tolerated and had a greater systemic antitumor response (in uninjected and visceral lesions) as compared to single agent. Ipilimumab + Laherparepvec (*n* = 98): The median duration of treatment with laherparepved and ipilimumab was 21.1 and 9.1 weeks, respectively. The median followup time and time to response was 68 weeks (Range: 0–156 weeks) and 5.8 months (95% CI: 5.4–10.9 months) respectively. 39% of the patients had an ORR (13% CR and 26% PR) and visceral lesions decrease was observed in 52% of the patient population. The median PFS was 8.2 months (95% CI: 4.2–21.5 months). Ipilimumab alone (*n* = 100): The median duration of treatment of ipilumumab was 9.1 weeks. The median followup time was 58 weeks (Range: 0–152 weeks). The median time to response was not estimated (HR = 1.41; 95% CI: 0.8–2.5). 18% of the patients had an ORR (7% CR and 11% PR; OR: 2.9, 95% CI: 1.5–5.5, *p* = 0.002) and visceral lesion decrease was observed in 23% of the patient population. The median PFS was 6.4 months (95% CI: 4.2–21.5 months).	Fatigue, chills, GI disorders, pruritus, rash and nausea.
Ipilimumab in Stage IV melanoma patients receiving palliative radiation therapy [149]	Active, not recruiting	Stanford University	II, NCT01449279	The combination was safe and efficacious. 22 patients were enrolled and treated in the study. 50% of the patients had benefited from the therapy including CR and PR at 55 week follow-up. 27.3% of the patients had an ongoing systemic complete response to the combination (95% CI- 9.7–56.9%) with no evidence of disease at 55 week. 27.3% (95% CI- 9.7–56.96%) of the patients has an initial PR without progression for median of 40 weeks (Range: 29–53). The median PFS was 26 weeks (Range: 2–65; 95% CI: 16.3–35.7) and the median overall survival was 55 weeks (Range: 8–141; 95% CI: 39.2–70.8) with the patients receiving the combination. The median time for response for patients who had CR or PR was 19 weeks (Range: 12–52). The strong antitumor response in patients with CR or PR can be attributed to increased levels of IL-2 producing CD8+ T cells and central memory CD8+ T cells in comparison to patients with melanoma could be used as biomarkers further.	Colitis, hypophysitis, rash, anemia, nausea and radiation dermatitis.
Ipilimumab alone or in combination with sargramostim (GM-CSF) in Stage III/IV melanoma that cannot be removed surgically [150]	Active, not recruiting	NCI	II, NCT01134614	The combination was advantageous and had a lower toxicity profile. The median follow-up was 13.3 months (Range: 0.03–19.9 months). Ipilimumab + Sargarmostim (*n* = 132): The median OS was 17.5 months (95% CI: 14.9-not reached). The one year survival rate was 68.9% (95% CI: 60.6–85.5%). Ipilimumab alone (*n* = 122): The median OS was 12.7 months (95% CI: 10-not reached). The one year survival rate was 52.9% (95% CI: 43.6–62.2%). There was no difference in PFS across both the groups.	GI toxicity, pulmonary toxicities,
Ipilimumab as adjuvant therapy after complete resection of high risk stage III melanoma [151]	Active, not recruiting	BMS	III, NCT00636168	Addition of Ipilimumab as adjuvant therapy benefited patients with microscopic involvement only (sentinel node-positive) and for patient with macroscopic or palpable nodes. The median follow up was 5.3 years. Ipilimumab group (*n*= 475): The 5 years recurrence-free survival and OS was 40.8% and 65.4% respectively. The rate of distant metastasis free survival at 5 years was 48.3%. Placebo group (*n* = 476): The 5 years recurrence free survival and OS was 30.3% (HR: 0.76; 95% CI: 0.64–0.89; *p* < 0.001) and 54.4% (HR: 0.72; 95% CI: 0.58–0.88; *p* = 0.001) respectively. The rate of distant metastasis free survival at 5 years was 38.9% (HR: 0.76; 95% CI: 0.64–0.92; *p* = 0.002).	Immune related adverse events- GI, hepatic, endocrine, skin and neurologic.
Combination of Anti-PD-1/PD-L1 and anti-CTLA-4					
Ipilimumab in low dose as an adjuvant in combination with nivolumab after resection of melanoma macrometastases [152]	Completed	Universitair Ziekenhuis Brussel	Ib, NCT02941744	Ipilumumab at low doses in combination with nivolumab had an acceptable safety profile. Ipilumumab (50 mg) + Nivolumab (10 mg) (*n* = 34): the median follow up was 86 weeks. One year relapse-free survival, overall survival and distant metastasis-free survival was 55% (95% CI: 39–72%); 97% (95% CI: 94–100%) and 79% (95% CI: 65–92%) respectively. Median relapse free survival was 84 weeks (95% CI: 28–139 weeks). Nivolumab (10 mg) (*n* = 22): The median follow up was 36 weeks. One year relapse-free survival and overall survival was 78% (95% CI: 73–82%) and 100% respectively. Distant metastasis was not observed. The median relapse free survival was not reached.	4–8% grade 3 immune related adverse events were across both the cohorts.
LTX-315 alone or in combination with Ipilimumab or Pembrolizumab in patients with transdermally accessible tumors [153,154]	Completed	Lytix Biopharma AS in collaboration with Theradex and ICON plcI	I, NCT01986326	Combination of immune checkpoint inhibitors with LTX-315 was safe and tolerable and demonstrated a potent anti-tumor activity. Of 6 melanoma patients received LTX-315 in combination with Ipilimumab, stable disease was observed in 33% of the patients. LTX-315 when administered to patients with solid tumors resulted in increase in number of CD8+ T cells at the site of treated lesions along with tumor infiltrating lymphocyte population. Clonal expansion of T-cells in blood was observed after treatment with LTX-315 as revealed by T-cell receptor sequencing.	LTX-315-related grade 3 and 4 adverse events (allergic/anaphylaxis) were observed along with tingling post injection, rash, fatigue, diarrhea, hypo and hyper tension, weakness.
Nivolumab and Ipilimumab alone or in combination in patients with previously untreated unresectable or metastatic melanoma (CheckMate067) [155]	Active, not recruiting	BMS	III, NCT01844505	Combination of ipilimumab and nivolumab or nivolumab alone was superior over monotherapy with ipilimumab. No new toxic effects associated with chronic use of these therapies were observed. Nivolumab plus Ipilimumab group (*n* = 314): Median overall survival was more than 60 months. Hazard ratio for death versus Ipilimumab group was 0.52. 5 year overall survival rate was 52%. Nivolumab group (*n* = 316): Median overall survival was 36.9 months. Hazard ratio for death versus Ipilimumab group was 0.63. 5 year overall survival rate was 44%. Ipilimumab group (*n* = 315): Median overall survival was 19.9 months. 5 year overall survival rate was 26%.	
Ipilimumab in combination with Nivolumab in patients with unresectable Stage III/IV malignant melanoma [156]	Active, not recruiting	BMS in collaboration with Medarex and Ono Pharma USA Inc	Ib, NCT01024231	The combination of Ipilimumab and nivolumab had durable clinical activity in patients with advanced melanoma. 94 patients were enrolled in the study. At the target lesions, the mean reduction in tumor burden was around 64.7%. The median follow-up was 30.3 to 55 months while the median OS was not reached at 3 years. The median PFS was 6.2 months (95% CI: 3.2–11 months). The median duration of response was 22.3 months (95% CI: 13.8–25.8 months). The best overall response rate and objective response rate by modified WHO criteria were 19.1% and 41.5% (95% CI: 31.4–52.1%). The CR and PR rates were 22.3% and 16% respectively. The OS and median PFS rates at 1 year were 81% (95% CI: 71–87%) and 37% (95%CI: 27–47%) respectively. The OS and median PFS rates at 2 years were 72% (95% CI: 62–80%) and 28% (95% CI: 19–38%) respectively. The OS and median PFS at year 3 were 63% (95% CI: 52–72%) and 17% (95% CI: 8–29%) respectively.	Grade 3 and 4 immune related toxicities such as rash, diarrhea, increase in lipase, AST, ALT and amylase, rthralgia, colitis, were observed.
Nivolumab alone or in combination with ipilimumab in melanoma patients with brain metastases [157]	Active, not recruiting	Melanoma Institute Australia in collaboration with Melanoma and Skin Cancer Trials Limited and BMS	II, NCT02374242	The combination of Nivolumab and ipilimumab was active in melanoma brain metastases with a durable intracranial and extra cranial response. Cohort A (*n* = 36): Nivolumab (3 mg/kg) + Ipilumumab (3 mg/kg) Cohort B (*n* = 27): Nivolumab (3 mg/kg) Cohort C (*n* = 16): patients with brain metastases for whom local therapy failed/showing neurological symptoms/leptomeningeal disease were administered nivolumab (3 mg/kg). As per RECIST criteria, the median follow up was 17 months (IQR: 8–25 months). The overall survival at 6 months in cohorts A.B and c were 78% (95% CI: 65–94%), 68% (95% CI: 52–89%) and 44% (95% CI: 25–76% respectively. Intracranial response: The overall response in cohorts A, B and C was 46% (95% CI: 29–63), 20% (95% CI: 7–41%) and 6% (95% CI: 0–30%) respectively. The PFS at 6 months in cohorts A, B and C was 53% (95% CI: 38–73%), 25% (95% CI: 9–44%) and 13% (95% CI: 5–65%) respectively. Extra cranial response: the overall response in cohorts A, B and C was 57% (95% CI: 37–75%), 29% (95% CI: 11–52%) and 25% respectively. The PFS at 6 months in cohorts A, B and C were 51% (35–76%), 35% (95% CI: 19–64%) and 19% (95% CI: 5–65%) respectively.	Grade 1 and 2 treatment related adverse events were commonly observed such as skin, GI, endocrine, musculoskeletal, respiratory related and fatigue. Grade 3/4 related adverse events were infrequent.
Pembrolizumab in combination with reduced dose Ipilimumab or Pegylated Interferon Alfa-2b in patients with advanced melanoma (KEYNOTE-29) [158]	Active, not recruiting	Merck Sharp & Dohme Corp.	I/II, NCT02089685	While the combination of Pembrolizumab and ipilimumab had good antitumor activity and manageable safety profile, the combination of pembrolizumab and PEF-INF did not. Pembrolizumab and Ipilimumab (*n* = 12): The median follow-up was 25.1 months (Range: 0.8–38.7 months). The median duration of response was not reached. The objective response rate as per independent central review was found to be 42% (95% CI: 15–72%). The CR and PR rates were 8.33% and 33.33% respectively. As per investigator review, the objective response rate and PR were 33% (95% CI: 10–655) and 33.33% respectively. Pembrolizumab and PEG-IFN (*n* = 17): The median follow up was 22.2 months (Range- 25–377 months). As per central and investigator review, the objective response rate was 20% and the partial response rate was 20%.	Pembrolizumab and Ipilimumab: Grade 1/2 treatment related adverse events- fatigue, diarrhea, rash, nausea, colitis, increased lipase, ALT and AST. Immune-related adverse events- colitis, hyper and hypo thyroidism. Pembrolizumab and PEG-IFN: Treatment related AEs- elevation is AST, ALT nerve disorder, fatigue, chills, pyrexia, diarrhea, rash, pruritus, nausea, anemia. Immune-related AEs- hyperthyroidism, pneumonitis, hepatitis.
Nivolumab combined with Ipilimumab or Ipilimumab alone in patients with untreated, unresectableor metastatic melanoma [159]	Active, not recruiting	BMS	II, NCT01927419	The median follow-up was 24.5 months (IQR: 9.1–25.7 months). Ipilimumab alone (*n* = 47): The 2 years overall survival was 53.6% (95% CI: 38.1–66.8%). The objective response rate and PR were 11% (95% CI: 3–23%) and 11% respectively. The median PFS was 3 months (95% CI: 27–5.1 months; HR: 0.36; 95% CI for HR: 0.22–0.56, *p* < 0.0001). The PFS and 1 and 2 year was 16% (95% CI: 6.6–28.9) and 152% (95% CI: 3.8–25.2%) respectively. Nivolumab plus Ipilimumab (*n* = 95): The 2 years overall survival was 63.8% (95% CI: 53.3–72.6%). The objective response rate, CR and PR were 59% (95% CI: 48–69%), 22% and 37% respectively. The median PFS was not reached. The PFS at 1 and 2 years was 52.5% (95% CI: 41.6–62.3%) and 51.3% (95% CI: 40.4–61.2%) respectively. The median overall survival was not reached in either group (HR: 0.74, 95% CI 0.43–1.26; *p* = 0.26).	Colitis, Diarrhea, Hypophysitis, Pneumonitis, Anaemia, Hypothyrodism, increased ALT.
Nivolumab administered sequentially with Ipilimumab in subjects with advanced or metastatic melanoma (CheckMate064) [160]	Active, not recruiting	BMS	II, NCT01783938	Nivolumab followed by ipilimumab was more clinically beneficial. Nivolumab followed by Ipilimumab (*n* = 70): The median overall survival was not reached. The 1 year overall survial was 76% (95% CI: 64–85%). The overall response rate, CR and PR was 56% (95% CI: 43.3–67%), 12% and 44% respectively. Ipilimumab followed by Nivolumab (*n* = 70): The median overall survival was 16.9 months (95% CI: 9.2–26.5 months; HR: 0.48; 95% CI for HR: 0.29–0.8). The 1 year overall survival was 54% (95% CI: 42–65%). The overall response rate, CR and PR was 31% (95% CI: 20.9–43.6%), 6% and 26% respectively.	Pruiritus, rash, fatigue, chills, pyrexia, vitiligo, diarrhoea, nausea, increased ALT, AST and lipase.

**Table 2 cancers-12-00482-t002:** Summary of clinical trials, outcomes and adverse events associated with novel BRAF inhibitors and ERK inhibitors in patients with metastatic melanoma.

Treatment	Status	Sponsor	Phase and NCT	Clinical Outcomes	Adverse Events
BRAF inhibitors					
LGX818 in patients with advanced or metastatic BRAF mutant melanoma [222]	Active, not recruiting	Array BioPharma	I, NCT01436656	25 BRAF-naïve and 29 BRAF inhibitor pretreated patients were enrolled in the study. The treatment was tolerable up to the MTD of 450 mg once daily, however the RP2D was declared as 300 mg once daily due to the increased risk of adverse events at 450 mg. BRAFi-naïve patients treated with 300–450 mg once daily saw an RR of 60% and PFS of 12.4 months (95% CI: 7.4-Not Reached), while for BRAFi-pretreated patients the RR was 22% and the PFS was 1.9 months (95% CI: 0.9–3.7 months).	Nausea, myalgia, PPED
LGX818 in combination with MEK162 in patients with advanced solid tumors [223]	Active, not recruiting	Array BioPharma	Ib/II, NCT01543698	The combination was safe with no substantial adverse evets observed. Nine BRAF naïve and 14 BRAF inhibitor pretreated patients were enrolled. The MTD was unable to be determined and the RP2D was 450–600 mg LGX818 + 45 mg MEK162 orally once daily. CR and PR for BRAF-naïve patients were 11% and 78% respectively. The PR for BRAF inhibitor pretreated patient groups was 21%.	Nausea, abdominal pain, headache, diarrhea, fatigue, visual impairment.
Encorafenib in combination with Binimetinib or Vemurafenib in patients with BRAF-mutant melanoma (COLUMBUS) [224]	Active, not recruiting	Array BioPharma	III, NCT01909453	The combination was efficacious, safe and tolerable. The median follow up and followup for overall survival was 32.1 months (95% CI: 29.5–32.3 months) and 36.8 months (95% CI: 35.9–37.5 months) respectively. Encorafenib + Binimetinib (*n*= 192): The median overall survival was 33.6 months (95% CI: 24.4–39.2 months. One and two year OS was 75.5% (95% CI: 68.8–81%) and 57.6% (95%CI: 50.3–64.3%) respectively. The median PFS was 14.9 months (95% CI: 0.2–2 months). PFS was longer in this group as compared to vemurafenib only group (HR:0.51, 95% CI: 0.39–0.67, *p* < 0.0001) and encorafenib only group (HR: 0.77, 95% CI: 0.59–1, *p* = 0.05). Overall response rates by masked independent central reiew was 64%. Vemurafenib only (*n* = 191): The median overall survival was 16.9 months (95% CI:14–24.5 months). Hazard ratio compared to combination group was 0.61 (95% CI: 0.47–0.79, *p* < 0.0001). The overall survival did not differ significantly. One and two year OS was 63.1% (95% CI: 55.7–69.6%) and 43.2% (95% CI: 35.9–50.2%) respectively. The median PFS was 7.3 months (95% CI: 5.6–7.9 months). Overall response rates by masked independent central reiew was 41%. Encorafenib only (*n*= 194): Overall survival was longer in this group compared to Vemurafenib only group (HR: 0.81, 95% CI: 0.58–0.98, *p* = 0.033). One and two year OS was 74.6% (95% CI: 67.6–80.3%) and 49.1% (95% CI:41.5–56.2%) respectively. The median PFS was 9.6 months (95% CI: 7.4–14.8 months). Overall response rates by masked independent central review was 52%.	Palmar-plantar erythrodysaesthesia, nausea, diarrhea, vomiting, fatigue, myalgia, arthralgia, increased γ-glutamyltransferase, increased blood creatine phosphokinase, hypertension
ERK inhibitor					
Ulixertinib in patients with advanced solid tumors [211]	Completed	BioMed Valley Discoveries, Inc.	I, NCT01781429	135 patients were enrolled in the study. The treatment was well tolerated and had an acceptable safety profile at doses of 600 mg twice daily and this dose was established as MTD and RP2D. This dose demonstrated anti-tumor activity in patients with treatment naïve or those progressed on MAPK pathway inhibitors. The PR was 12%.	Diarrhea, fatigue, dehydration, nausea, rash, dermatitis acneiform, increased blood creatinine.

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
