# Peer review of "Current Advances in the Treatment of BRAF-Mutant Melanoma"

_cancers, 2020, doi:10.3390/cancers12020482_

Round 1
Reviewer 1 Report
Page 1
L12: 5-y-OS can be as high as 26-52% if you look at the data of Checkmate-067, KEYNOTE-006 or COMBI-v/d
L17: encorafenib
L17: 'sequential or stand-alone therapies'. This is unclear.
L23: BRAF
L34: 5-y-OS can be as high as 26-52% if you look at the data of Checkmate-067, KEYNOTE-006 or COMBI-v/d. Of course this is a selected population, but worth mentioning.
Page 2
L64: I'd add inclusion in a clinical trial
L70: and/or: There is currently no approval for combined targeted therapy and immune checkpoint inhibition
L81: 'sequential or stand-alone therapies'. This is unclear.
L86: no anti-PDL1: there are no anti-PDL1 approvals for melanoma
Page 3
L96: BRAF
L100: vemurafenib
L100: has
L137: Q61, not Q16
Page 4
L165: You should specify that PD-1 and CTLA4 are localized on T-cells.
L166-167: it has also been shown that combined BRAFi/MEKi renders the tumor microenvironment more immunoresponsive (Ascierto, Oncoimmunology 2018), hence the rationale to combine anti-PD1 and BRAFi/MEKi in clinical trials
L171: There are trials ongoing with PI3K and CDK4/6 inhibitors in combination with BRAF/MEK-inhibitors. You should also mention them.
Page 5
L178: Spartalizumab is a PD-1 blocking antibody that is being investigated in COMBI-i (DAB/TRA + SPARTA/PLB)
L182: combination with ipilimumab
Page 7:
L248: these trials already have been done. Hodi et al NEJM.
Page 8:
L268: I'd split this table in one with completed trials and another one with trials that are still recruiting.
I would focus on trials in advanced melanoma only
Some important trials have not been included in the table such as the KEYNOTE-002, KEYNOTE-006 and COMBI-i
Page 21
L324: Why don't you discuss dabrafenib or vemurafenib?
L325: Preferentially V600E, but also others V600 isoforms
Why don't you discuss MEK-inhibitors?
Page 23
L376:
Why don't you discuss MEK-inhibitors or BRAF/MEK-inhibitors? MEK-inhibitors delay resistance to BRAF-inhibitors. They are the current standard of care for BRAF-mutant melanoma, your topic of review.
Page 25:
L408: together

Reviewer 2 Report
This review is focused on treatment of BRAF mutant melanoma with targeted or immune therapies. It gives a general view of treatment options that are available for this subtype of melanoma. The authors provided a complete and detailed table with outcomes and adverse events in current clinical trials involving anti PD-1/L1 and CTLA4 and ERK inhibitors.
Major points
The author should cite and discuss seminal papers that are not included:Regarding BRAFi and MEKi resistance: PMID: 25600339; PMID: 23569304; PMID: 24265153; PMID: 26359985; PMID: 22763439
Regarding targeted and immune therapies cross resistance: PMID: 24577748; PMID: 27846054; PMID: 29449059
Regarding ERKi: PMID: 28714990
Nowadays, in addiction to immune check point inhibitors (ICB), there are other type immune therapies available, such as TRL9 agonists, oncolytic virus or neoantigen vaccination, and their efficacy in combination with ICB is under evaluation in different clinical trials. The authors should discuss also these type of therapies. Maybe this is out of the scope of the review, but the authors do not discuss at all about resistance to ICB
Minor points
There are some spelling errors on BRAF (line 23 and line 96) Table 1 is very useful, but sometime very busy (especially the clinical outcomes), a better separation between each clinical trial will be helpful (maybe with horizontal lines) The author are using B-RAF and BRAF, they should be consistent and using only one nomenclature
Round 2
Reviewer 2 Report
The paper have been improved and now is good for publication